# Reversible single crystal-to-single crystal double [2+2] cycloaddition induces multifunctional photo-mechano-electrochemical properties in framework materials

Dylan A. Sherman [1], Ryuichi Murase[1], Samuel G. Duyker [1], Qinyi Gu [1], William Lewis [2], Teng Lu[3], Yun Liu [3] & Deanna M. D'Alessandro [1✉]

Reversible structural transformations of porous coordination frameworks in response to external stimuli such as light, electrical potential, guest inclusion or pressure, amongst others, have been the subject of intense interest for applications in sensing, switching and molecular separations. Here we report a coordination framework based on an electroactive tetra-thiafulvalene exhibiting a reversible single crystal-to-single crystal double $[2+2]$ photo-cyclisation, leading to profound differences in the electrochemical, optical and mechanical properties of the material upon light irradiation. Electrochemical and in situ spectro-electrochemical measurements, in combination with in situ light-irradiated Raman spectroscopy and atomic force microscopy, revealed the variable mechanical properties of the framework that were supported using Density Functional Theory calculations. The reversible structural transformation points towards a plethora of potential applications for coordination frameworks in photo-mechanical and photoelectrochemical devices, such as light-driven actuators and photo-valves for targeted drug delivery.

[1] School of Chemistry, The University of Sydney, Sydney, NSW 2006, Australia. [2] Sydney Analytical, Core Research Facilities, The University of Sydney, Sydney, NSW 2006, Australia. [3] Research School of Chemistry, The Australian National University, Canberra 2601, Australia. ✉email: deanna.dalessandro@sydney.edu.au

Of the numerous nanoporous materials reported to date, only a handful of examples explore light as a stimulus to influence the material's chemical, electrochemical, optical, and mechanical properties. Rarer still is the combination of light modulated activity with two or more of these interrelated properties in a given material. Coordination frameworks (a.k.a. metal-organic frameworks (MOFs)) provide an ideal platform for multifunctionality. Owing to the well-defined 3D nature of these crystalline materials, they exhibit nanoscale porosity and highly tunable chemistry which is brought about through the judicious selection of metal nodes and organic ligand building blocks[1,2]. Many thousands of coordination frameworks have been investigated for use in gas storage and separation processes, whilst others have been explored for their catalytic and semiconducting properties[3].

An interesting avenue to engender multifunctionality in coordination frameworks is the [2 + 2] photocyclisation reaction, a process that is widely used for the synthesis of natural products, cage compounds and polymers[4]. Photo-inducing [2 + 2] cycloaddition of olefins to form cyclobutane is a versatile organic reaction requiring only UV irradiation. Schmidt first studied the solid-state reaction in cinnamic acids in 1964, establishing two necessary geometric conditions[5,6]. First, the double bonds must be oriented in parallel, and second, the bonds must be within a proximal distance of 3.5–4.2 Å to maximise $p_z$-orbital overlap[7–13]. While a limited number of [2 + 2] photocycloadditions in framework materials have been observed[12], only a handful of these occur in a single crystal-to-single crystal (SC-SC) manner[14], which is essential for retaining framework structural stability. Although single crystal X-ray diffraction (SCXRD) is clearly suited to identifying SC–SC transformations, reports are limited due to the challenges associated with retaining single crystallinity in the presence of such dramatic structural transformations and the associated stress generation[15].

Thus, [2 + 2] photocycloadditions in coordination frameworks are still relatively rare, and the overwhelming majority of examples employ the 1,2-bis(4′-pyridyl)ethylene (bpe) ligand and its derivatives rather than more complex heteroatomic ligands[14,16]. Photocyclisation reactions have primarily been utilised to extend the dimensionality of framework materials. For example, Vittal and co-workers have reported examples of creating higher dimensionality polymers by linking 0D, 1D or 2D networks together[17,18]. Although not strictly a coordination framework, an interesting case is the transformation of a molecule, fantrip, to a 2-D polymer framework of poly(fantrip) which occurs via SC–SC [4 + 4] cycloaddition[19]. Only a few publications to date report SC–SC photocyclisation in 3D frameworks, and none have been structurally porous[13]. Work from the Vittal group demonstrated the first example of a 3D to 3D [2 + 2] cycloaddition in a framework[20,21]. One 3D coordination polymer exhibits unprecedented double SC–SC olefinic photocyclisation by stacking 1,4-bis[2-(4-pyridyl)ethenyl]benzene (1,4-bped) ligands in-phase using a Cd metal centre[22]. More recent work by Lang and co-workers has extended [2 + 2] photocyclisation beyond structural applications for frameworks to control fluorescence switching in 3D frameworks[23,24]. Reversibility of [2 + 2] cyclisation in the reported examples remains a challenge and only a few have successfully demonstrated the ability to return to their original form in an SC–SC manner[25–27]. Such a feature is a necessary prerequisite for applications in photo-switches and photosensors which rely on reversible photo-mechanical energy transformation.

A prospect yet to be considered is the concept of employing SC–SC photocyclisation to introduce functionality such as reversible switching and property modulation. In this regard, [2 + 2] photocyclisation potentially adds another dimension of utility to alter the inherent chemistry of coordination frameworks, including their innate electrochemical, optical and mechanical properties. To do so, however, requires extending [2 + 2] photocyclisation beyond bpe based ligands in frameworks, by judiciously designing and selecting heteroatomic ligands that enable framework properties such as magnetism, electroactivity or conductivity. In implementing this strategy, our study selects an electroactive tetrathiafulvalene (TTF)-based ligand that has not been used before, offering prospects for multifunctional redox-active material design.

Here we describe a reversible double [2 + 2] photocyclisation of an electroactive TTF core, and an example of a coordination framework exhibiting a reversible SC–SC double [2 + 2] photocyclisation leading to profound differences in the electrochemical, optical and mechanical properties of the material upon light irradiation. The photocyclisation reaction is facilitated by the cofacial arrangement of the TTF ligands in the framework, where the double bonds are oriented in parallel and within a distance sufficient to maximise $p_z$-orbital overlap. Such an arrangement optimises the conditions required for the SC–SC cyclisation as outlined by Schmidt[28], and facilitates through-space charge transfer interactions of the nature shown previously in cofacial MOFs[29,30]. The framework structure $[Cd_2(Py_2TTF)_2(bpdc)_2]\cdot$solvate (1) {$Py_2TTF = 2,6$-bis($4′$-pyridyl)-tetrathiafulvalene; $bpdc^{2-} = 4,4′$-biphenyldicarboxylate} based on the redox-active $Py_2TTF$ ligand was found to undergo a SC–SC transformation to $[Cd_2(Py_4C_{12}S_8H_4)(bpdc)_2]\cdot$solvate (3) via double [2 + 2] photocyclisation of the TTF core that was reversible by heating. Additionally, a partially cyclised intermediate structure, $[Cd_2(Py_4C_{12}S_8H_4)_{0.84}\{(Py_2TTF)_2\}_{0.32}(bpdc)_2]$ (2), was obtained. Light-irradiated PXRD and Raman spectroscopy were employed to monitor the structural transformation in situ with the aid of Density Functional Theory (DFT) calculations to support our interpretation. Furthermore, a rigid framework of $[Cd_2(Py_2TTF)(bdc)_2]$ (4) ($bdc^{2-} = 1,4$-benzenedicarboxylate) which precludes the photocyclisation demonstrates the structural and energetic conditions required for the occurrence of the transformation between 1 and 3. The propensity for 1 to undergo a SC–SC [2 + 2] photocyclisation, coupled with the virtue of possessing a redox-active ligand, demonstrates for the first time modulation of an electroactive MOF stimulated via light. The combination of tuneable optical, electrochemical and spectro-electrochemical (SEC) properties in 1 and 3 adds another element of versatility that can be altered in conjunction with the structural transformation. Lastly, light-irradiated atomic force microscopy (AFM) studies were conducted on 1 to reveal the variable mechanical properties of the framework. Our ongoing investigation of the reversible structural change of the TTF core points towards a plethora of potential applications including photo-valves for targeted drug delivery and in multifunctional photo-mechano-electrochemical switching devices, amongst others.

## Results

**Synthesis and crystallography.** Framework 1 was prepared by the solvothermal reaction of $Cd(NO_3)_2\cdot4H_2O$ with $Py_2TTF$ and $H_2bpdc$ in $N,N′$-dimethylformamide/ethanol (DMF-EtOH) by heating a solution of the components to 80 °C. Bright orange plate-like crystals of $[Cd_2(Py_2TTF)_2(bpdc)_2]$ (1) were formed over a period of 4 days and structural determination was conducted by single crystal X-ray diffraction (Fig. 1 and Supplementary Fig. 1). The material crystallises in the monoclinic space group, $P2_1/n$, with unit cell parameters $a = 10.1851(4)$ Å, $b = 28.1899(11)$ Å, $c = 14.9875(5)$ Å, $\beta = 97.675°$ and $V = 4264.6(3)$ Å$^3$. All crystallographic data are provided in Supplementary Table 1.

The key structural feature of 1 is the presence of cofacially aligned pairs of $Py_2TTF$ ligands that pillar the undulating 2D sheets of $\{Cd(bpdc)\}_n$, which propagate in the $a$-$b$ direction (Fig. 1a and Supplementary Fig. 1). Closer inspection of the eight-

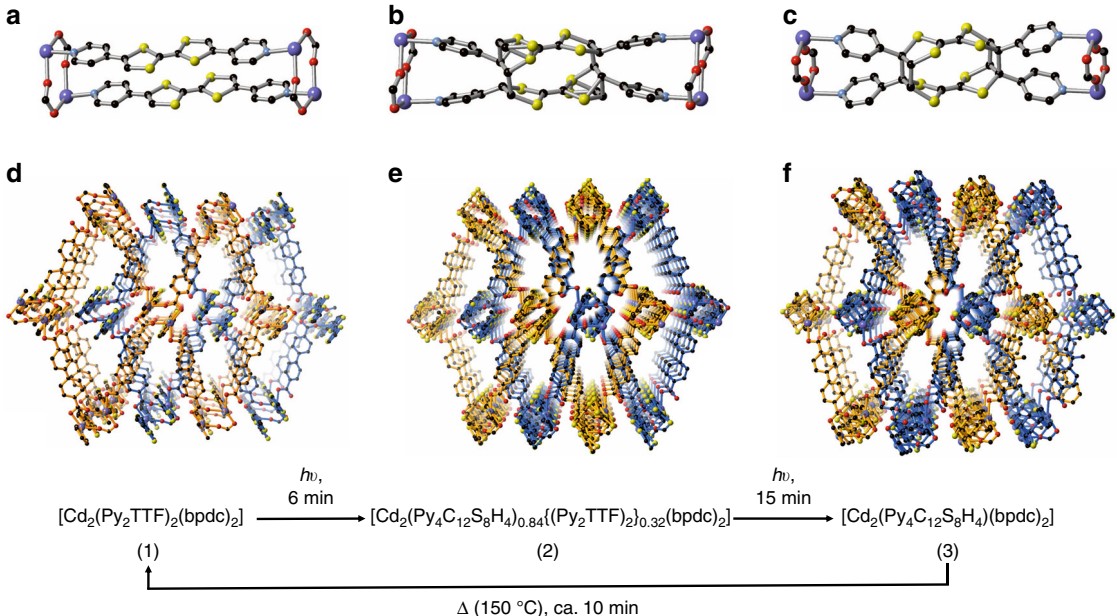

**Fig. 1 Effect of light irradiation on the structure of 1.** The cofacially arranged $Py_2TTF$ moieties of (**a**) **1**, (**b**) **2** and (**c**) **3** experience an unprecedented double [2 + 2] photocyclisation reaction. The view down the *c*-axis of (**d**) **1**, (**e**) **2** and (**f**) **3**. The two independent nets have been highlighted in orange and blue. The coloured spheres represent C (black), N (light blue), O (red), S (yellow) and Cd (violet). Hydrogen atoms and solvent molecules in each of the frameworks have been omitted for clarity.

membered $\{(Cd–O–C–O)_2\}$ ring formed by two of the Cd(II) centres and the bpdc units reveals the Cd⋯Cd separation to be only 3.869 Å despite the large cation size. The TTF-based dimers adopt a face-to-face π-stacking orientation with a slight translational shift that staggers the ligands, aligning corresponding S atoms of each TTF core with a proximal S⋯S distance of 3.77 Å. The large rectangular cavities that run in all three directions afford a framework that occupies approximately 28% of the unit cell volume. Thus, a second interpenetrating framework, independent of the first, is found, translated approximately mid-way along the unit cell length (Fig. 1d). Despite the presence of this second network, the framework of **1** retains open channels along the *c*-direction (42.7% void volume, based on the Van der Waals surface).

In view of the structure of **1**, we envisioned that a [2 + 2] photocycloaddition was possible, as the orientation and proximity of the $Py_2TTF$ units in the structure fulfilled the criteria outlined by Schmidt[28]. Our first indication that a structural transformation was successfully induced was based on the colour change of the crystals of **1** when exposed to natural light over a period of four days. The crystals of **1** turned from bright orange to light yellow in colour to generate the structure of $[Cd_2(Py_4C_{12}S_8H_4)(bpdc)_2]$ (**3**; Fig. 1e). Initial attempts to reproduce this effect under simulated sunlight using a desk lamp or UV light emitting diodes (LEDs) for a prolonged time (>1 h) led to the loss of single crystal character. However, we were able to generate crystals of suitable quality by exposing crystals of **1** to a 25 W microscope lamp for 15 min. Examination of this structure (**3**) revealed a crystal system reminiscent of **1** which possesses the same monoclinic space group, $P2_1/n$, with unit cell parameters of $a = 9.113(2)$ Å, $b = 28.336(7)$ Å, $c = 15.948(4)$ Å and $\beta = 95.629(5)°$. This translates to an overall decrease in the unit cell volume by 161.8 Å³.

Remarkably, the structure of **3** revealed a significant structural transformation of **1** at the cofacial $Py_2TTF$ ligands of independent nets, whereby the original olefinic bonds that cap each pair of sulfur atoms around the C=C bond in the TTF core had cyclised to form cyclobutane rings, chemically bonding the previously distinct $Py_2TTF$ dimer units together (Fig. 1f).

Despite the numerous examples of TTF-based complexes and framework materials that fulfil these criteria, only five examples of complexes have been crystallographically shown to undergo photo-dimerisation, all of which occur on only one side of the TTF core[31–35]. The ability for this material to undergo double photocyclisation likely arises from the framework providing geometrical constraints to fulfill Schmidt's criteria, while also exhibiting flexibility to respond to the structural change. The once planar $Py_2TTF$ ligands now also exhibit puckering at the S atoms and the pyridyl groups adopt a V-like conformation to retain coordination to the Cd(II) centers. Distortion of the 8-membered Secondary Building Unit (SBU) is also apparent whereby the Cd⋯Cd separation is lengthened significantly to 4.126 Å. With these structural changes, the pore apertures of **3** are also slightly modified, with the channels that run along the *c*-direction experiencing a contraction (Fig. 1d–e). Crystals of **3** showed superior thermal stability compared with **1** as evidenced by thermogravimetric analysis (TGA) which showed sharp mass loss upon heating above ca. 300 °C in comparison to ca. 275 °C in **1**, and were stable in air indefinitely (Supplementary Figs. 2 and 3).

**Probing the photocyclisation process.** SCXRD has been used to study the mechanistic details of a number of chemical reactions, however such investigations are relatively rare in framework chemistry[36]. We envisioned that conversion of **1** to **3** could be controlled by varying the light exposure of the crystals. By exposing crystals of **1** to 6 min of light from a 100 W halogen lamp source we were able to generate and structurally characterise a partially cyclised form of **1**, $[Cd_2(Py_4C_{12}S_8H_4)_{0.84}\{(Py_2TTF)_2\}_{0.32}(bpdc)_2]$·DMF (**2**) using synchrotron radiation. The framework of **2** still retains the symmetry of **1** and the unit cell parameters were found to lie between that of **1** and **3** (Table 1). In examining the structure of **2**, we found that the $Py_2TTF$ ligand had undergone an incomplete [2 + 2] cycloaddition reaction with half of the $Py_2TTF$ exhibiting disorder. As depicted in Fig. 1b, the disorder alludes to 84% conversion

**Table 1 Summary of unit cell parameters of framework 1 and frameworks 2 and 3 obtained upon light-irradiation.**

| Compound | 1 | 2 | 3 |
|---|---|---|---|
| Exposure time | 0 min | 6 min | >15 min |
| $a$ (Å) | 10.1851(4) | 9.2500(19) | 9.113(2) |
| $b$ (Å) | 28.1899(11) | 28.297(6) | 28.366(7) |
| $c$ (Å) | 14.9875(5) | 15.864(3) | 15.948(4) |
| $\beta$ (°) | 97.675(3) | 95.75(3) | 95.629(5) |
| Volume (Å$^3$) | 4264.6(3) | 4131.4(15) | 4102.8(18) |

(calculated from the refined site occupancies) to the dimerised ($Py_4C_{12}S_8H_4$) form with the remaining 16% existing as the original $Py_2TTF$ form. Attempts to capture reaction intermediates of shorter exposure proved difficult due to both the fragile nature of the crystals and the shorter time scale of the experiment.

To provide further insight into the structural and energetic requirements for the transformation between **1** and **3**, a model framework of $[Cd_2(Py_2TTF)(bdc)_2]$ (**4**) was synthesised using the more rigid 1,4-benzenedicarboxylate co-ligand and similar reaction conditions to that for **1**. The most significant difference between the structures is the orientation of the $Py_2TTF$ dimers, which were found to exist in a herringbone stacking arrangement in **4** (Supplementary Figs. 4 and 5). The staggered arrangement of these ligands leads to a significantly distant intermolecular separation between the TTF units (C=C, 4.46–4.74 Å). Light irradiation of the bulk material using both sunlight and a 25 W light source showed no evidence of a structural transformation. In relation to Schmidt's criteria[7,8], framework **4** significantly exceeds the geometric proximity required for photocyclisation to occur.

In addition to our single crystal study, the in situ conversion of the bulk powder of **1** was confirmed by light-irradiated powder X-ray diffraction and Raman studies. A powdered sample of **1** in a slurry of DMF-EtOH was exposed to light from twelve blue LEDs (0.28 W and 51 lumens each; 300–600 nm spectral range) and the evolution of the powder patterns was monitored over a 17.5-h period at 300 K. A comparison of the calculated powder patterns for **1** and **3** revealed that the most significant Bragg reflections are observed in the $2\theta = 5$–20° range (Fig. 2a and Supplementary Figs. 6–8). The experimental patterns collected at room temperature show slight deviation in the position of the peaks in contrast to the calculated plot due to unit cell expansion. As shown in Fig. 2a, the most prominent difference is observed for the 010 peak which shifts in $2\theta$ from 6.8° to ca. 6.5° upon irradiation, becoming a shoulder to the 020 reflection. A second significant shift is that of the 10−1 peak from $2\theta = 9.6°$ to 10.2°, becoming more intense. These changes correlate well with the expected powder pattern of **3**. Additionally, the complete regress of the 011 peak is indicative of quantitative generation of **3**.

The Raman spectra of TTF and its derivatives also possess characteristic features that often enable the oxidation state to be elucidated. In our case, due to the retention of the crystal symmetry in **1** and **3** and the photocyclisation being confined to the $Py_2TTF$ moieties, Raman spectroscopy provided an ideal insight into the structural transformation. A spectrum was collected using 785 nm laser excitation with 30-s intervals of light irradiation between spectral collections over the course of 10.5 min. Additionally, DFT calculations were undertaken to model the vibrational spectra. Our initial trials to model the fragments that incorporated both the $\{Cd_2(bpdc)_4\}$ SBU and the $Py_2TTF/(Py_4C_{12}S_8H_4)$ moieties were not successful. However, satisfactory calculated spectra were obtained when precluding the SBU to analyse just the $Py_2TTF$ and the cyclised ($Py_4C_{12}S_8H_4$) fragments of **1** and **3**, respectively (Supplementary Figs. 9 and 10).

A summary of the Raman active vibrations and their assignments is provided in Supplementary Tables 2 and 3, and Supplementary Scheme 1.

By comparing the experimental single point spectra of **1** and **3** with the calculated spectra, all vibrational modes associated with the pyridyl rings of the ligand in the computational spectra are shifted to higher energies (ca. 50–100 cm$^{-1}$). These variations were expected and are caused by the absence of cadmium ions in our model. As shown in Fig. 2b, a number of distinct changes were observed in the Raman spectra upon light irradiation. Firstly, the regress of four peaks at 831, 847, 940 and 1527 cm$^{-1}$ was observed. These peaks are assigned to the C–S stretching ($\nu_3$), C–C stretching ($\nu_7$), C–S stretching ($\nu_6$) and pyridyl ring wagging, respectively in reference to our computational study. The growth of new peaks at 735, 783, 1078 and 1559 cm$^{-1}$ corresponding to the stretching of the newly formed C–C bond ($\nu_7$), C–S stretching ($\nu_3$), pyridyl ring bending and pyridyl ring wagging, respectively, were also observed. Overall, the spectral changes are consistent with the transformation of cofacially aligned $Py_2TTF$ ligands in **1** to the cyclised ($Py_4C_{12}S_8H_4$) form in **3**. Notably, the growth in the peaks at 783 and 1559 cm$^{-1}$ is the most significant indication of the photocyclisation reaction. To quantitatively monitor the completeness of the cyclisation reaction, the 1527 cm$^{-1}$ peak proved to be the most suitable given its full regression upon completion. By deconvoluting this peak, it was found that almost full conversion (ca. 98.6%) of **1** to **3** was achieved over 540 s of light irradiation (Supplementary Table 4).

**Reversibility.** Photocyclisation reactions are not usually reversible because they require the breaking of bonds to reform olefins[25,26]. Our successful in situ study of the structural transformation of **1** to **3** prompted a further study to explore the reversibility of the reaction (Supplementary Figs. 11 and 12). At room temperature, both in ambient light and when kept in the dark, samples of **3** remained stable and did not exhibit any retro-conversion to the original state. Remarkably, heating crystals of **3** as a slurry in DMF in dark conditions at 150 °C for 48 h saw a complete retro-conversion of all crystals to the cofacial **1** structure. The conversion was visually observable, with a clear change in the crystal colour from the yellow **3** to the dark orange colour of **1**. The retro-conversion was confirmed by PXRD of the heated sample which revealed a diffraction pattern corresponding to **1** (data for the converted and retroconverted forms is provided in Supplementary Fig. 13). A sample of **1** was also cycled through two complete structural switches (**1**-**3**-**1**-**3**-**1**), with PXRD data showing no loss of crystallinity after each conversion. After the second retro-conversion the sample retained an ability to cyclise without loss of crystallinity for a third time.

Due to the reduced high temperature stability of **1** compared with **3** and the longer timescale required for the retro-conversion, in situ monitoring was not feasible via XRD methods. However, the reverse process was successfully studied in situ using Raman spectroscopy (785 nm). Heating (180 °C) and concurrent monitoring of the crystals of **3** yielded the spectral changes shown in Fig. 2c over a period of ca. 4.5 min. These changes correspond well to the light-irradiated in situ Raman spectroscopy discussed earlier and the final spectrum which resembles that of **1**. Using the same approach to the photo-irradiated Raman experiment, deconvolution of the 1527 cm$^{-1}$ peak enabled evaluation of the retro-conversion, and full conversion was observed over 270 s of light irradiation (Supplementary Table 5). Note that this method is semi-quantitative with an estimated uncertainty limit based on the accuracy with which band areas could be calculated of ±5%. In both the PXRD and Raman methods, regeneration of **1** required the samples to be wetted with DMF as the absence of the

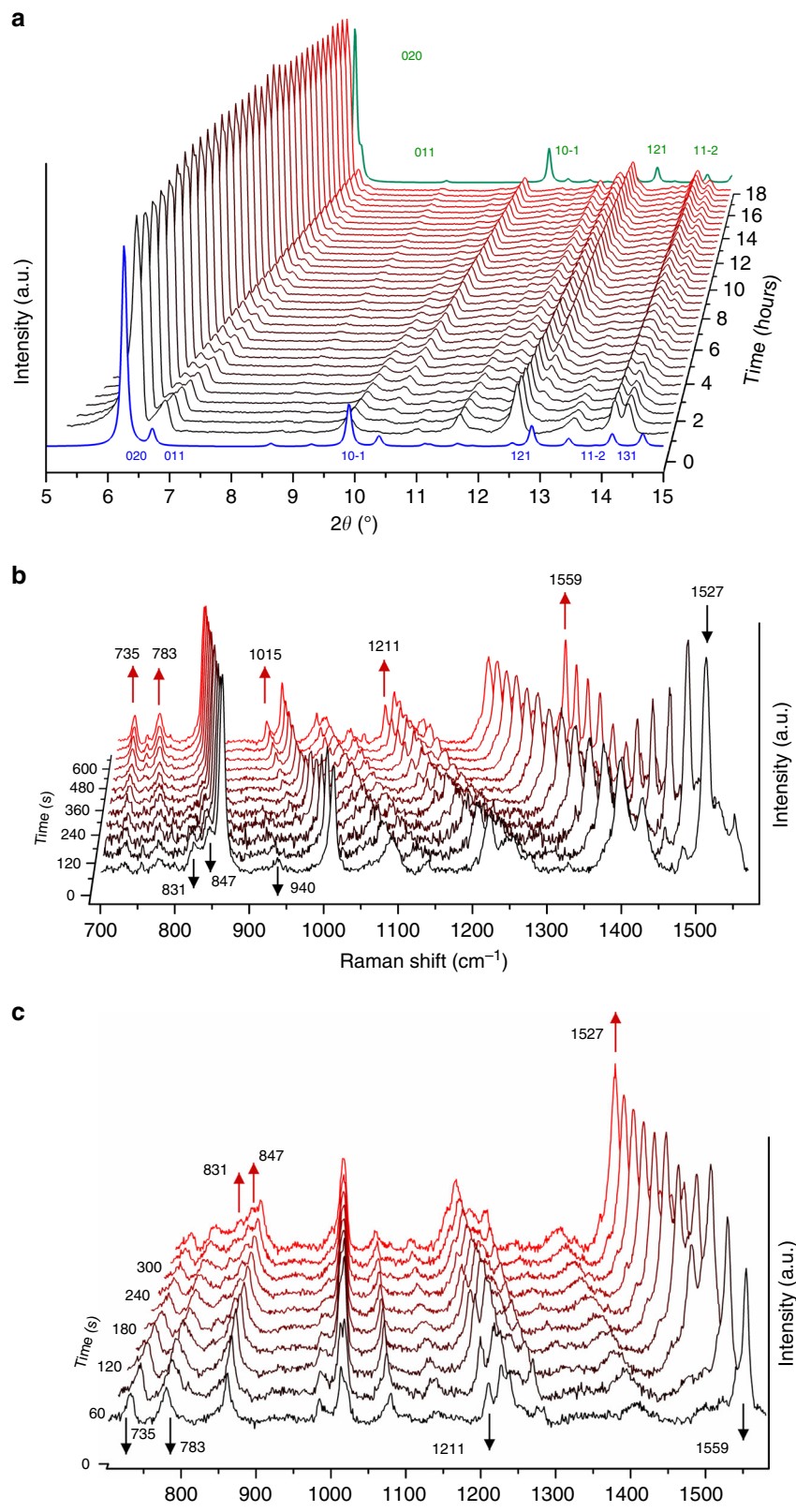

**Fig. 2 Evidence for the conversion of 1 to 3 and the reversible retro-conversion. a** Light-irradiated PXRD of **1** over a period of 17 h. The starting pattern is shown in black (front) and the last pattern is shown in red (back). The blue and green patterns are the simulated powder patterns of **1** and **3**, respectively. The key *hkl* reflections have been labelled. **b** Light-irradiated Raman (785 nm) of **1**. The first and the last spectra are shown in black (0 s) and red (630 s), respectively. **c** Isothermal Raman (785 nm) of **3** at 180 °C. The first and the last spectra are shown in black (0 s) and red (660 s), respectively.

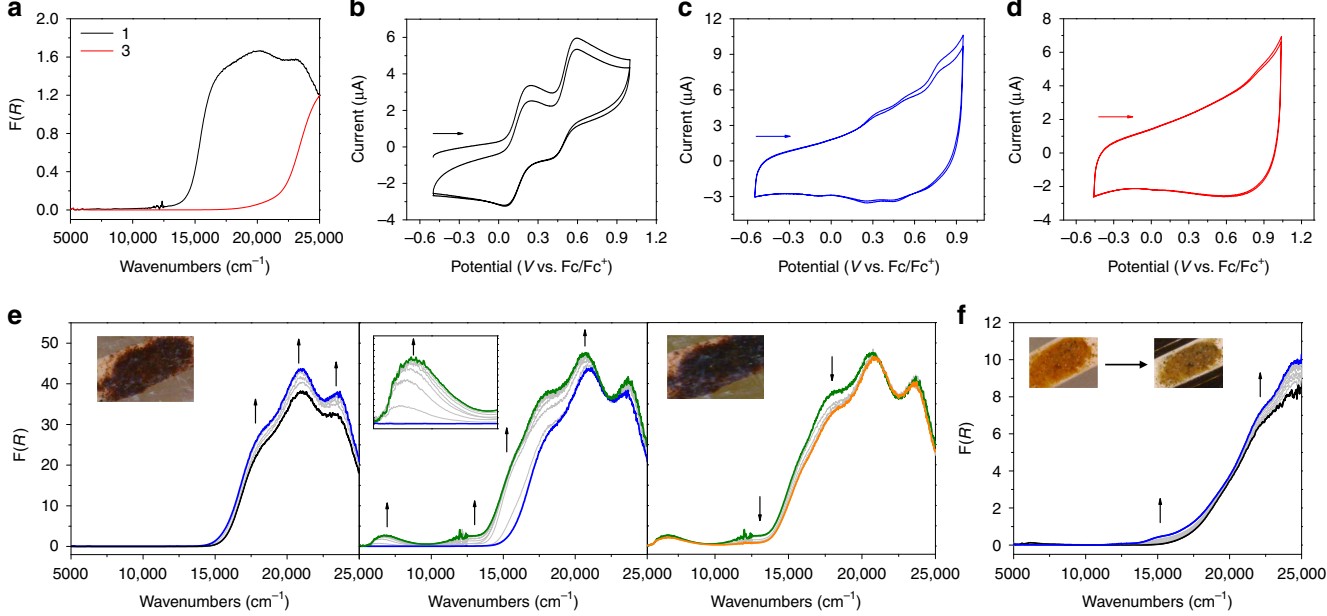

**Fig. 3 Photocyclisation dependent electrochemical and spectral data. a** Diffuse reflectance spectrum of **1** (black) and **3** (red). Cyclic voltammogram of (**b**) **1**, (**c**) **1** after exposure to UV lamp (20 W) for 30 minutes, and (**d**) **3** in 0.1 M [(*n*-C$_4$H$_9$)$_4$N]PF$_6$/CH$_3$CN recorded at 200 mV s$^{-1}$. Arrows indicate the direction of the forward scan. (**e**) Vis-NIR SEC of **1** in 0.1 M [(*n*-C$_4$H$_9$)$_4$N]PF$_6$/CH$_3$CN between applied potentials of +0.3 to +0.6V (left), +0.6 to +0.8V (middle) and +0.8 to +0.9 V (right). Inset figures show the colour of the crystals at 0.0 V (left) and +0.8 V (right). (**f**) Vis/NIR SEC of **3** between applied potentials of 0.0 to +0.7 V. Inset—starting sample of **3** at 0.0 V (left) and the sample at +0.7 V (right). Arrows indicate the direction of the spectral progression.

solvent led to loss of crystallinity. Although the conditions required for the reverse process indeed presents challenges for its applicability, the reversible nature of the photocyclisation from **3** to **1** provides this material with the additional rare virtue of repeatable 'switchability'.

**Modulation of electrochemical and optical properties.** Structural transformations of framework materials have been investigated towards a number of applications; common examples are frameworks that can alter their pore apertures in situ or via post-synthetic methods, which have seen great promise for their tuneable gas sorption/separation properties[37–40]. In our case, the virtue of having a redox-active ligand that undergoes an electronic rearrangement within the framework proposes another dimension of tuneability that can be controlled, especially when coupled with variations in mechanical and structural properties. To better understand the physical property changes that accompany the SC–SC transformation, the solid state spectroscopic, electrochemical, spectroelectrochemical and mechanical properties of **1** and **3** were probed.

The optical characteristics of framework materials can provide insight into their fundamental electrochemical properties. Bulk samples of **1** and **3** retain the stark colour changes observed in single crystals, with **1** existing as a dark red microcrystalline powder, while **3** is a light yellow powder. As shown in Fig. 3a, three transitions are observed in the diffuse reflectance spectrum of the as-synthesised framework of **1**. The bands at 20120 and 23310 cm$^{-1}$ are assigned to a π–π* and an intramolecular charge transfer transition (ICT) of the Py$_2$TTF ligand, respectively, as indicated from previous studies[41]. The lower energy band at 17400 cm$^{-1}$ is attributed to the small presence of radical Py$_2$TTF$^{·+}$ in the framework. Similar observations have been made in both discrete complexes and frameworks that employ TTF-based ligands[42]. In studying the diffuse reflectance spectrum of **3**, a stark contrast in the spectra of the two frameworks is found with **3** exhibiting an overall blue-shift of the spectrum. DFT calculations for molecular orbitals of the cofacial and

cyclised fragments of **1** and **3** revealed that the HOMO–LUMO band gap in **1** (1.106 eV) is significantly smaller than that in **3** (2.195 eV). Additionally, the two transitions that were present at 17,400 and 20,120 cm$^{-1}$ are completely diminished in **3**. This is consistent with the structural changes associated with the Py$_2$TTF becoming a cyclised (Py$_4$C$_{12}$S$_8$H$_4$) dimer which no longer possesses the characteristic electrochemical properties for TTF.

Ligands incorporating TTF have been shown to inherit the redox properties of the core in numerous examples[42]. Py$_2$TTF itself possesses favourable electroactivity with accessible and stable oxidation states[43,44]. Investigating the solid state cyclic voltammogram of **1** revealed two *quasi*-reversible one electron processes reminiscent of the electrochemistry of Py$_2$TTF (Fig. 3b and Supplementary Fig. 14). Scanning anodically, at 400 mV s$^{-1}$ the first process at 0.34 V (vs. Fc/Fc$^+$) is attributed to the generation of the Py$_2$TTF$^{·+}$ radical. This is followed by the second oxidation to Py$_2$TTF$^{2+}$ at 0.69 V (vs. Fc/Fc$^+$). In contrast to the traditional electrochemistry of **1**, the cyclic voltammogram of **3** (Fig. 3d) reveals only one irreversible oxidation at 0.83 V (vs. Fc/Fc$^+$). The cyclic voltammogram of a sample of **3** retro-converted to **1** (Supplementary Fig S15) shows that the electrochemical properties of the material return to those of the the as-synthesised sample of **1**, demonstrating an ability to switch the material's redox activity by the reversible double [2 + 2] photocyclisation.

To further elucidate the transition, the cyclic voltammogram of an intermediate stage was measured by exposing a sample of **1** to a UV lamp (20 W) for 30 min in 0.1 M [(*n*-C$_4$H$_9$)$_4$N]PF$_6$/CH$_3$CN (Fig. 3c). The voltammogram at 400 mV s$^{-1}$ reveals a more convoluted series of processes when scanning anodically. Owing to the large capacitance, a further square wave voltammetry (SWV) study was undertaken which revealed three oxidative and two reductive processes in the reverse scan (Supplementary Fig. 16). Based on the cyclic voltammogram of Py$_2$TTF and the position of these faradaic processes, the first and third oxidative processes at 0.26 and 0.52 V (vs. Fc/Fc$^+$), respectively, can be attributed to Py$_2$TTF. This likely arises from a small proportion of

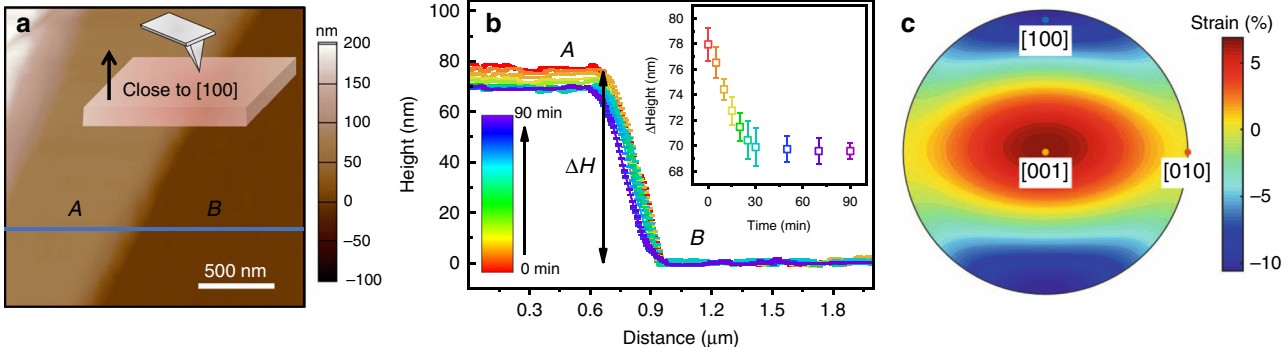

**Fig. 4 In situ light irradiated AFM data in a single crystal. a** The morphology of the sample **1** scanned over the region 2 μm × 2 μm where the schematic drawing in the insert presents the location of the AFM tip on the crystalline surface with a possible crystallographic orientation close to [100]; **b** Line profiles scanned under different illumination times across the crystal terraces of the A and B regions, corresponding to the blue solid line in (**a**). The inset in (**b**) shows the height difference of the A and B regions as a function of exposure time. (3) The strain tensor induced by the SC-SC phase transition from **1** to **3**.

the uncyclised cofacial form still present in the framework, as the removal of conjugation from the TTF core in **3** reduces the ability for electron delocalisation, which stabilises the oxidation states accessible to $Py_2TTF$. The assignment of the irreversible oxidation at 0.83 V (vs. $Fc/Fc^+$) cannot definitively be identified, but the process is consistent with the reported electrochemical behaviour of a one-sided [2 + 2] photocyclised TTF based molecule[31]. This intermediate state, however, does demonstrate that the degree of cyclisation of the material can be used to control the material's electrochemical properties.

Spectroelectrochemistry (SEC) is a powerful method applied to study the optical properties of materials whilst modulating the redox states in situ[45,46] and has been utilised in framework chemistry to identify both fundamental phenomena such as the generation of partial charge states and mixed valency, in addition to its more practical application in devices such as electrochromic windows and solar cell panels[47,48]. A solid state Vis-NIR SEC study of **1** revealed three major processes upon oxidation (Fig. 3e). The first at 0.3–0.6 V (vs. $Ag/Ag^+$) involves the formation of $Py_2TTF^{\cdot+}$ radical as indicated by the increase of the associated band at 18400 $cm^{-1}$ in addition to the two other bands in the Vis/UV region. At +0.7 V, a more complex process is observed with the key feature being the formation of a low energy NIR band which is assigned to a through-space Intervalence Charge Transfer (IVCT) transition between the cofacial $Py_2TTF$ moieties in **1** that possess partially oxidised mixed-valence $TTF^0$/ $TTF^{\cdot+}$ cores, akin to the similar processes characterised in closely related frameworks containing cofacially aligned electroactive ligands[29,30]. A final process occurs between 0.8 and 0.9 V corresponding to $Py_2TTF^{2+}$ formation. Additionally, the previously red crystals of **1** had obtained a darker brown colour (Fig. 3e, insets). Partial regeneration of the starting spectrum was observed when a cathodic potential of −0.5 V (vs. $Ag/Ag^+$) was applied, indicating that the rate of reduction may be significantly slower than that of the oxidation process.

In the related structure of **4** which precludes the cofacial arrangement of $Py_2TTF$ ligands found in **1** and adopts a significantly larger intermolecular distance between the ligands, both the electrochemical and Vis-NIR data lend support to our interpretation (Supplementary Figs. 17 and 18). The $Py_2TTF$ ligands in **4** behave as isolated, non-interacting units with no evidence for the IVCT interaction found in **1**.

The Vis-NIR SEC of **3** in 0.1 M [($n$-$C_4H_9$)$_4$N]$PF_6$/$CH_3CN$ revealed significantly distinct properties to those observed for **1**. Only one process occurred upon applying an anodic potential between +0.3 and 0.7 V (vs. $Ag/Ag^+$) (Fig. 3f). During this process a new broad band arises between ca. 13,000–16,000 $cm^{-1}$,

whilst the ICT transition present in the neutral state increases in intensity at 24,000 $cm^{-1}$. The process was characterised by a colour change from orange to brown (Fig. 3f, inset). A slight intensification of the band at 22,100 $cm^{-1}$ is also observable and is tentatively assigned to a metal-to-ligand charge transfer (MLCT) transition. Together, these analyses provide no definitive profile of the redox behaviour exhibited by **3**.

**Photo-active mechanical behaviour.** To understand the mechanical property variations between **1** and **3** that accompany the optical and electrochemical property changes, we examined photo-induced strain behaviour by using AFM for in situ characterisation under light illumination. A crystal of **1** was first immersed in DMF under a dark environment. The surface of the sample **1** was step-shaped with large flat crystal terraces caused by different cleavage planes with the same crystallographic orientation (Fig. 4a insert). Figure 4a shows the typical sample morphology scanned over a 2 μm × 2 μm area, where the two flat regions (labelled by "A" and "B") neighbour one another, and exhibit a height difference i.e., Δh of ~78 nm. The time required for the phase transition depends on the intensity of the light. As it takes around 15 minutes to complete the phase transition from **1** to **3** under a normal microscope lamp, the same area was repeatedly measured with a designed scan rate of 2 Hz (i.e. ~2 min per image) under illumination using the AFM. The time-resolved line profiles are plotted in Fig. 4b with the height of region B set as a zero point for comparison. The height difference between the regions A and B linearly decreases when the sample is initially exposed to light and then remains unchanged after ~30 min, suggesting the accomplishment of a photo-inducing SC−SC phase transition from **1** to **3** under the given experimental conditions (see the inset in Fig. 4b).

When the SC−SC phase transition occurs, the strain change in the regions A and B should be equivalent, i.e., $\Delta l_A/l_A = \Delta l_B/l_B$ ($l_A > l_B$). Thus, the decrease in the height difference between the crystal terraces A and B suggests shrinkage along this normal direction during the photo-induced phase transition from **1** to **3**. According to the crystallographic information listed in Table 1, the sample does not present cell-multiplicity during the SC−SC phase transition. Therefore, the strain induced by such a phase transition can be estimated based on the crystal structure[49] and Fig. 4c is the resultant strain tensor induced by the phase transition from **1** to **3**. The orientation around the $a$-axis of **1**, i.e., close to the [100] direction, presents a large shrinkage (strain ~ −11%) and the [010] direction has the maximum expansion. The crystalline direction away from [100] and [010] will make the strain gradually smaller. The photo-induced strain observed from

the AFM measurement can be roughly estimated as $(\Delta l_A\text{-}\Delta l_B)/\Delta h \approx -10.25\%$, which is in good agreement with the phase transition strain along the $a$-axis. These results suggest that the normal direction of the measured plane is close to the crystallographic [100] direction.

Such large photo-induced strain observed in the liquid environment is significant, presenting a promising potential to develop various functional devices used in biological and/or liquid environments, such as photo-valves and photo-active displacement controllers for drug delivery to remotely control drug release or control the rate of liquid flow. Most importantly, this photomechanical behaviour, together with abovementioned photochromic, electrochromic, and photoelectrochemical effects derived from the SC−SC phase transition, provides a possibility to design a multifunctional device that enables a combination of the functionalities of a sensor, transducer and actuator. Further device fabrication is beyond the scope of this paper and will be the focus of our future research efforts.

## Discussion

The photo-mechano-electrochemical properties of **1** provide a new paradigm for multifunctional materials design. The reversible double $[2+2]$ photocyclisation phenomenon observed for the electroactive TTFs within the structure is facilitated by the cofacial alignment of the ligands, which are separated by 3.77 Å, fulfilling Schmidt's criteria[28]. Electrochemical measurements in addition to a suite of solid state in situ spectroscopic methods, including Vis/NIR and Raman techniques, reveal the profound changes in the electrochemical and optical properties of the material upon light irradiation. Meanwhile, in situ light irradiated AFM confirmed the mechanical transformation of the framework upon photocyclisation of the ligands and revealed large photo-induced strain which could further complement the electrochemical and optical property variations. Beyond these switchable properties, other functions may be controllable through ligand-photocyclisation, including the modulation of gas sorption properties.

By extending double $[2+2]$ photocyclisation in MOFs to heteroatomic ligand sites, this study uncovers the potential for a plethora of framework properties, derived from and controlled by ligand design, that can be modulated. As TTF is also redox-active, the double $[2+2]$ photocyclisation becomes a unique mechanism that can repeatedly switch the electroactive state of the framework.

The prospect of using natural light−an almost unlimited resource accessible to all−to stimulate the conversion of **1** to **3**, and consequently control the framework's optical, electrochemical and mechanical properties, creates exciting prospects for the use of these materials in light-driven separations processes and as solar-driven actuators or photo-thermal-valves for drug delivery, amongst others. The design parameters for photoactive frameworks elucidated here serve as a blue print for a new generation of multifunctional photo-electroactive frameworks, into which current studies are underway in our laboratory.

## Methods

**Single crystal X-ray diffraction**. Measurements of single crystal X-ray diffraction data for **1** and **4** were undertaken on an Oxford Supernova diffractometer with Cu–$K\alpha$ radiation at 100 K. Diffraction data for **2** was collected on the MX1 beamline at the Australian Synchrotron with 0.71073 Å radiation at 100 K. Diffraction data for **3** was collected on the Bruker ApexII FR591 diffractometer with Mo–$K\alpha$ radiation at 100 K. Crystals were extracted from the mother liquor into paratone oil and mounted onto the goniometer for data collection. In the case of **1**, the crystals were cold-mounted under dry ice to prevent the degradation of the crystals. The structures were solved using SHELXT[50] and refined using a full-matrix least squares procedure based upon $F^2$ using SHELXL[51]. Structure solution and refinement was performed within the WinGX[52] system of programs and OLEX2[53]. Crystal information and details relating to the structural refinements are presented in Supplementary Table 1 and in Supplementary Note 1. Data are available from

the Cambridge Crystallographic Database as numbers CCDC 1898207-1898210 (**1–4**, respectively).

**Powder X-ray diffraction**. Powdered samples were loaded in a 0.5 mm diameter capillary and sealed. For **1**, the sample was loaded into the capillary as a slurry in DMF. Single point measurements were performed over the 5–50° 2θ range with a 0.02° step size and 2° min$^{-1}$ scan rate on a PANalytical X'Pert Pro diffractometer fitted with a solid-state PIXcel detector (40 kV, 30 mA, 1° divergence and anti-scatter slits, and 0.3 mm receiver and detector slits using Cu-Kα ($\lambda$ = 1.5406 Å) radiation.

**Light irradiated powder X-ray diffraction**. Light-irradiated PXRD analysis was undertaken on a PANalytical X'Pert Pro diffractometer fitted with a solid-state PIXcel detector (40 kV, 30 mA, 1° divergence and anti-scatter slits, and 0.3 mm receiver and detector slits using Cu-Kα ($\lambda$ = 1.5406 Å) radiation. A powdered sample of **1** in a slurry of DMF was loaded into a 0.3 mm diameter capillary and $4 \times 3$ blue LED ($12 \times 0.28$ W) strips were mounted around the capillary. Single point measurements were taken at 30 min intervals with continuous light irradiation onto the sample.

**Electrochemistry and spectroscopy**. Cyclic voltammograms were collected on a BASi Epsilon electrochemical analyser. All measurements were recorded in 0.1 M $[(n\text{-}C_4H_9)_4N]PF_6/CH_3CN$ electrolyte using a glassy carbon electrode, platinum counter electrode and an Ag reference electrode. All potentials are reported in volts versus Fc/Fc$^+$ couple. UV-Vis-NIR spectra were collected on an Agilent CARY5000 Spectrometer with a Harrick Omni-Diff probe. Spectra were collected between 5000 and 25,000 cm$^{-1}$ and are reported as the Kubelka–Munk transform, where $F(R) = (1 − R)^2/2R$.

**Solid-state electrochemistry**. Solid state electrochemical measurements were performed using a Basi Epsilon electrochemical analyser. Argon was bubbled through solutions of 0.1 M $[(n\text{-}C_4H_9)_4N]PF_6/CH_3CN$. The CVs were recorded using a glassy carbon working electrode (1.5 mm diameter), a platinum wire auxiliary electrode and an Ag reference electrode. The sample was mounted on the glassy carbon working electrode by dipping the electrode into a paste made of the powder sample in the supporting electrolyte.

**Solid-State UV-Vis-NIR spectroscopy**. UV-Vis-NIR spectra were obtained on powdered samples at room temperature using an Agilent CARY5000 Spectrophotometer equipped with a Harrick Omni-Diff Probe accessory over the wavenumber range 5000–25000 cm$^{-1}$. Spectra are reported as the Kubelka–Munk transform, where $F(R) = (1 − R)^2/2R$.

**Solid-state Vis-NIR spectroelectrochemistry**. In the solid state, the diffuse reflectance spectra of the electrogenerated species were collected in situ in a 0.1 M $[(n\text{-}C_4H_9)_4N]PF_6/CH_3CN$ electrolyte over the range 5000–25,000 cm$^{-1}$ using a Harrick Omni Diff Probe attachment and a custom built solid state spectroelectrochemical cell[45]. The cell consisted of a Pt wire counter electrode and an Ag/AgCl reference electrode in 3 M NaCl aqueous solution. The solid sample was immobilised onto a 0.1 mm thick indium tin oxide (ITO) coated glass slide (which acted as the working electrode) using a thin strip of Teflon tape. The applied potential was controlled using an eDAQ e-corder 410 potentiostat. Continuous scans of the sample were taken on the CARY5000 spectrometer and the potential increased gradually until a change in the spectrum was observed. Spectra are reported as the Kubelka–Munk transform, where $F(R) = (1 − R)2/2R$.

**Raman spectroscopy**. Single point Raman spectra were measured using an inVia Renishaw Confocal Raman microscope. The laser (785 nm) was focused onto the sample using the Raman microscope (×50 magnification). The Raman spectra were recorded over the 100–3200 cm$^{-1}$ range with 10 seconds exposure time and 10% laser power over 1 accumulation.

**Light-irradiated Raman spectroscopy**. Light irradiated Raman spectra were measured using an inVia Qontor Confocal Raman microscope. The laser (785 nm) was focused onto freshly prepared crystals of **1** which were wetted with DMF using the sample stage microscope (×50 magnification). The Raman spectra were recorded over the 600–1700 cm$^{-1}$ range with 1 s exposure time and 5% laser power over 30 accumulations. The light power of the white light from the microscope was measured by a light metre and was 12.45 kFc (=200.4 watts or 134.0 kLumens). Spectra were collected between 30 s intervals of light irradiation. The light from the sample stage cavity and room were turned off to ensure no other source of light was irradiating the sample during the experiment.

**Isothermal Raman spectroscopy**. Isothermal Raman spectra were measured using an inVia Qontor Confocal Raman microscope. The laser (785 nm) was focused onto single crystals of **3** using the sample stage microscope (×50 magnification). Details of the data range, exposure time, laser power and number of accumulations

are identical to those reported above. The samples were placed onto a Linkam Scientific FTIR 600 heating stage and the temperature was set by an external controller. Upon heating the sample, each spectrum was collected in 30-s intervals.

**Evaluation of the structural transformation**. The percentage conversion of framework from **1** to **3** was calculated by monitoring the peak centred at 1527 cm$^{-1}$ in in the light irradiated Raman spectra. The conversion to **3** was deemed complete when full recession or no further change in the 1527 cm$^{-1}$ peak was observed. Deconvolution of the peaks yielded the integrated area which was then used to calculate the percentage conversion as a fraction r relative to the starting spectrum of **1**. Spectral deconvolutions were undertaken in OriginPro software using the Lorentzian model with fixed baselines. The modelled peak positions for each spectrum were adjusted to give the best possible fit with the maximum deviation being 4 cm$^{-1}$.

The same procedure was used to quantify the conversion of **3** to **1** from the isothermal Raman experiment. In this case, the percentage conversion was calculated as a fraction relative to the spectrum of **3**. A summary of the parameters is provided in Supplementary Tables 4 and 5.

**Atomic force microscopy (AFM)**. The in situ morphology mapping was conducted by the Asylum Research Cypher Atomic Force Microscopy (Cypher AFM) equipped with the droplet liquid cell. A single crystal sample of **1** was placed on the flat Si-substrate and a drop (~100 μL) of DMF-EtOH was used to immerse the single crystal for preventing the decomposition. The AFM tip (Olympus AC240TM) with the calibrated spring constant, $k = 2.07$ N/m (resonance peak ~70 kHz in the air while ~10 kHz in the DMF), was used to conduct the non-contact imaging under the illumination induced by the built-in optical microscopy.

**DFT computational calculations**. DFT calculations on the Py$_2$TTF dimer in its cofacial and cyclized forms were performed using DMol3 within Materials Studio. Geometry optimisations and energy calculations were carried out using the 'fine' overall quality setting, with the Perdew–Wang (PWC) local density approximation (LDA) functional[54], Ortmann, Bechstedt and Schmidt (OBS) correction for dispersion forces, and a smearing value of 0.005 Hartrees to treat partial orbital occupancies. The geometry of the cyclised dimer was optimized while the cofacial geometry was fixed to that determined in the single crystal structure. HOMO and LUMO energies and wavefunctions were calculated, as well as three-dimensional maps of electron density and electrostatic potential.

Additional vibrational frequency calculations were performed using Gaussian 16[55]. The M05-2X functional and 6-311 + G(2df,p) basis set were used. Geometries of both the cyclised dimer and individual neutral Py$_2$TTF ligand were optimized, with the former adopting a conformation close to that found in the crystal structure. The ligand optimized to a non-planar geometry with bending at the sulfur atoms, consistent with the unbound, neutral molecule.

**Syntheses**. 2,6-Bis(4′-pyridyl)-tetrathiafulvalene (Py$_2$TTF) was synthesised according to a modified literature procedure as detailed in the Supplementary Methods[56].

**Synthesis of [Cd$_2$(Py$_2$TTF)$_2$(bpdc)$_2$]·5DMF (1)**. Cd(NO$_3$)$_2$·4H$_2$O (11.6 mg, 0.0375 mmol), Py$_2$TTF (13.5 mg, 0.0375 mmol) and biphenyl-4,4′-dicarboxylic acid (9.1 mg, 0.0375 mmol) were placed in a 21 mL solvothermal vial which was covered with aluminium foil and dissolved in 4 mL of DMF and 0.75 mL of EtOH. The mixture was heated to 130 °C for 10 min and further heated at 80 °C in a convection oven with no exposure to light, yielding red plate-like crystals after four days. Loss of crystallinity was observed upon exposure to air at room temperature over a period of ca. 12 h, and thus further characterisation of **1** was undertaken on samples wetted with DMF or acetonitrile (MeCN), or over a limited timeframe (< 2 h) to prevent structural degradation. Yield: 20 mg (30.4 % based on Cd(II)). IR (ATR) (cm$^{-1}$): 3058 (w), 2928 (w), 2852 (w), 1941 (w), 2468 (w), 1671 (s), 1605 (s), 1576 (s), 1524 (s), 1498 (w), 1393 (s), 1252 (w), 1224 (w), 1176 (m), 1094 (s), 1018 (w), 939 (w), 851 (m), 818 (w), 770 (m), 729 (w), 682 (m), 661 (m), 628 (m). Analysis calc. for C$_{77}$H$_{81}$Cd$_2$N$_9$O$_{13}$S$_8$ (%): C 50.46, H 4.48, N 6.92, S 14.08; Found (%): C 50.43, H 4.40, N 6.90, S 14.03.

**Synthesis of [Cd$_2$(Py$_4$C$_{12}$S$_8$H$_4$)$_{0.84}$(Py$_2$TTF)$_{0.32}$(bpdc)$_2$] (2)**. Samples of **2** for single crystal X-ray diffraction studies were generated from wetted crystals of **1** in DMF via light irradiation for 6 min using white light (25 W) from a microscope lamp. The partially cyclised product was then transferred immediately from the microscope slide to the diffractometer in paratone oil under dark conditions for analysis.

**Synthesis of [Cd$_2$(Py$_4$C$_{12}$S$_8$H$_4$)(bpdc)$_2$]·3DMF (3)**. Single crystals of **3** were formed by irradiating single crystals of **1** with white light (25 W) using the beam from a microscope. Bulk **3** was formed by placing a sample of **1** in direct sunlight for 2 days or under a UV lamp (20 W) for 1 h. IR (ATR) (cm$^{-1}$): 3054 (w), 2918 (w), 2847 (w), 1663 (s), 1605 (m), 1576 (m), 1522 (m), 1384 (s), 1254 (w), 1174 (w), 1090 (m), 1020 (w), 884 (w), 849 (m), 770 (s), 735 (w), 706 (w), 680 (m), 659 (m),

624 (m). Analysis calc. for C$_{71}$H$_{67}$Cd$_2$N$_7$O$_{11}$S$_8$ (%): C 50.89, H 4.03, N 5.85, S 15.31; Found (%): C 50.85, H 3.98, N 5.80, S 15.28.

**Synthesis of [Cd$_2$(Py$_2$TTF)$_2$(bdc)$_2$]·DMF (4)**. Cd(NO$_3$)$_2$·4H$_2$O (11.6 mg, 0.0375 mmol), Py$_2$TTF (13.5 mg, 0.0375 mmol) and 1,4-benzenedicarboxylate (6.4 mg, 0.0375 mmol) were dissolved in DMF (4 mL) and EtOH (0.75 mL). The mixture was sealed and after sonication for 5 min was heated in a solvothermal oven at 80 °C for 72 h to yield red plate-like crystals suitable for X-ray diffraction (11.2 mg, 11.3% based on Cd(II)). IR (ATR) (cm$^{-1}$): 3045 (w), 2930 (w), 2864 (w), 2485 (w), 1195 (w), 1663 (m), 1605 (m), 1549 (s), 1376 (s), 1219 (m), 1090 (s), 1015 (s), 941 (m), 824 (s), 748 (s), 690 (s), 659 (s), 626 (s). Analysis calc. for Cd$_2$C$_{57}$H$_{48}$N$_7$O$_{12}$S$_8$ (%): C 45.51, H 3.22, N 6.52, S 17.05; Found (%): C 45.49, H 3.15, N 6.45, S 17.08.

## Data availability

The data that support the findings of this study are available from the corresponding author upon request. The X-ray crystallographic data for structures reported in this study have been deposited at the Cambridge Crystallographic Data Centre (CCDC). CCDC numbers for **1–4** are CCDC 1898207-1898210. These data can be obtained free of charge from The Cambridge Crystallographic Data Centre via www.ccdc.cam.ac.uk/data_request/cif.

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

## Acknowledgements

The authors gratefully acknowledge support from the Australian Research Council (FT170100283, DP160104780), the Vibrational Spectroscopy Core Facility and Sydney Nano at the University of Sydney in addition to the Australian Synchrotron Access Program. Dr Peter Turner's assistance with preliminary X-ray crystallographic measurements is also gratefully acknowledged. This paper is dedicated to the memory of Emeritus Professor Noel S. Hush AO FRS FNAS FAA FRACI FRSN (1924-2019)

## Author contributions

D.A.S. undertook the synthetic work and the majority of physical and structural characterisation with significant input from R.M., in addition to R.M. and Q.G. who undertook additional experimental work. X-ray structural data analysis was undertaken by D.A.S., R.M. and W.L. Computational calculations were undertaken by S.D. AFM measurements and their analysis were undertaken by T.L. and Y.L. The overall project was directed and supervised by D.M.D. R.M. wrote the majority of the initial draft of the paper which was based on the thesis of D.A.S. All authors provided comments and input into the final version of the paper.

## Competing interests

The authors declare no competing interests.
