## [Peer Review File · Nature Communications]

Reviewers' comments:

Reviewer #1 (Remarks to the Author):

In this manuscript, the authors reported the first example of a coordination framework based on an electroactive tetrathiafulvalene exhibiting a reversible single crystal-to-single crystal double [2+2] photocyclisation leading to profound differences in the electrochemical, electronic, optical and mechanical properties of the material upon light irradiation. The experimental result indicated that the reversible structural transformation points towards a plethora of new potential applications for coordination frameworks in photo-mechanical and photoelectrochemical devices such as light-driven actuators and photo-valves for targeted drug delivery. I recommend its publication after the authors properly answer the following concerns.

1. For the elemental analysis of 4, its observed C content did not well match that of the theoretical value. Thus the authors should re-measure its elemental analysis.
2. The structural transformation of coordination polymers are evidenced by ¹H-NMR spectra. Thus the ¹H-NMR spectra of coordination polymer 1 before and after UV light irradiation should be provided.
3. In Figure S11, there are some flaws in the PXRD of coordination polymers. Thus the authors should re-measure their PXRD or perform the ¹H-NMR spectra to confirm the complete retro-conversion of these complexes.
4. Some interesting works related to the coordination polymers for single crystal-to-single crystal [2+2] photocyclisation should be cited. For examples: *Angew. Chem. Int. Ed.*, 2018, 57, 12696-12701; *Angew. Chem. Int. Ed.*, 2010, 49, 4767; *Chem. Commun.*, 2018, 54, 5831; *Chem. Commun.*, 2016, 52, 7990; *Inorg. Chem.*, 2018, 57, 849.
5. There are still some typo and grammatical mistakes in the manuscript which should be corrected and polished before its acceptance in this journal.

Reviewer #2 (Remarks to the Author):

In this work, the authors report a reversible single crystal-to-single crystal double [2+2] cycloaddition in a metal-organic framework (MOF) which consists of electroactive tetrathiafulvalene (TTF) monomers. The SC-SC photocyclization has been reported in the past for several examples of MOFs and 2D polymer frameworks. But the examples of realizing the reversible photocyclization remain limited. Thus, the major novelty of this work as claimed by the authors lies in the first example of a MOF exhibiting a reversible SC-SC double [2+2] photocyclization based on the TTF core, which also leads to varying the electrochemical and optical properties. Basically, the novelty of the work can be fine if there are interesting properties demonstrated for the SC-SC photocyclization process in the newly synthesized MOFs. Nevertheless, there are substantial issues required to be addressed before the further consideration of this work for a possible publication in *Nature Communications*:

-There are too many claims such as, "the first example of..." and "a novel example of..." in the context. The authors shall stay more neutral when describing major claims. I believe that every piece of work should be published as the first example of a specific research field.

-It is good that the authors use PXRD and in-situ Raman spectroscopy to demonstrate the reversibility of transition between 1 and 3. Nevertheless, the missing key information of this work is about the quantification of the reversibility. From the latter results of square wave voltammetry and spectroelectrochemistry, it is clear that not all the monomers undergo the [2+2] photocyclization. Since the complete conversion of monomers can influence the intrinsic properties of the MOFs, the quantification of conversion and reversed ring opening reaction shall be demonstrated. Moreover, how many times have the authors repeated the reaction cycles between 1 and 3?

-The authors claimed the modulation of electronic properties between 1 and 3, but to be frank, I could not really find the electronic property studies in the context. The authors mainly describe the optical or photophysical studies of bulks samples of 1 and 3. For the electronic property, I would expect some investigations on the electronic conductivity or so.

-The AFM investigation on MOF crystal phase change is interesting, but I am afraid that the connection

to mechanical behavior is not really strong. It is more relevant for the volume change or height change of the crystals. For the mechanical behavior or study, I would expect something else, like the pressure, tensile strength, etc.

-Some speculative sentences like in page 5 "We postulate that the double [2+2] photocycloaddition is possible...." shall be removed out.

-In the supplementary information, the reaction scheme for the synthesis of Py2TTF and the chemical structures of all intermediate compounds should be shown.

Reviewer #3 (Remarks to the Author):

In their manuscript, D'Allessandro and coworker present a tetrathiafulvalene-containing metal-organic framework containing which undergoes [2+2] photocyclization.

The MOF and its phase changes were carefully analyzed by x-ray diffraction and Raman and optical spectroscopy. Changes of the redox potential were shown by cyclovoltammetry. Atomic force microscopy showed a shrinking of the crystal as result of the phase change due to light irradiation. The data is well presented and comprehensible. The manuscript is well written.

Unfortunately, the field of light responsive MOFs is only partially cited and a classification of the present work is difficult. For instance, MOF single crystal-to-single crystal switching has been presented several times.

While the first MOF driven actuators have been published, here the potential application as actuator is given only as outlook. Likewise the electronic properties where MOFs with switchable conduction have been already presented. Both features go beyond the here presented work. So, it remains vague how the presented MOF performs in comparison to the published MOF materials.

Although the presented MOF is certainly interesting from a scientific point of view, I cannot spot the major breakthrough expected for a publication in a high impact journal.

Due to the lack of novelty, I cannot recommend the publication in Nat. Comm. and suggest a more specialized journal.

Response to Reviewers' comments for NCOMMS-19-11694-T

We thank the reviewers for their questions and have provided below a detailed point-by-point response below, in addition to highlighting changes in the manuscript.

Reviewer #1:

In this manuscript, the authors reported the first example of a coordination framework based on an electroactive tetrathiafulvalene exhibiting a reversible single crystal-to-single crystal double [2+2] photocyclisation leading to profound differences in the electrochemical, electronic, optical and mechanical properties of the material upon light irradiation. The experimental result indicated that the reversible structural transformation points towards a plethora of new potential applications for coordination frameworks in photo-mechanical and photoelectrochemical devices such as light-driven actuators and photo-valves for targeted drug delivery. I recommend its publication after the authors properly answer the following concerns.

1. For the elemental analysis of **4**, its observed C content did not well match that of the theoretical value. Thus the authors should re-measure its elemental analysis.

Response: We thank the reviewer for their constructive criticism and have re-measured the elemental analysis of framework **4**. It should be noted that it has been the case, in our experience, that the elemental analysis data for nanoporous MOF materials are variable between repeat measurements depending on the precise treatment conditions of the materials. For example, MOF samples maintained under atmospheric conditions often adsorb moisture, while volatile solvents can be liberated from the pores. In the present case, the re-measurement of the sample has provided data that are now within a more reasonable tolerance level of the theoretical values.

2. The structural transformation of coordination polymers are evidenced by $^1\text{H-NMR}$ spectra. Thus the $^1\text{H-NMR}$ spectra of coordination polymer **1** before and after UV light irradiation should be provided.

Response: We thank the reviewer for their recommendation. We do recognise that [2+2] photocyclisation reactions of framework materials are often monitored using $^1\text{H-NMR}$ and did consider this with primacy when conducting this study. It was possible to digest **1** using KOH and obtain a spectrum that was consistent with Py_2TTF and biphenyl dicarboxylate (top, left). However, framework **3** could not be digested. Various bases were tested as digestion agents, noting that acids are well recognised as protonating TTF cores, along with changing other conditions. The NMR data suggests that significant decomposition of **3** has occurred (top, right). We were unable to synthesise the cyclised TTF-dimer as a free molecule and theorise that once the framework is disrupted during digestion, the cyclised molecule does not remain stable and also decomposes.

In addition to this limitation, we also aimed to collect more definitive data that monitored the transition *in situ*, and not merely at the end-points of the transition *ex situ*. The structural transformation of the solid

coordination framework was therefore evidenced by X-ray crystallography and PXRD, coupled with Raman analysis to analyse local chemical environments in a similar way to $^1\text{H-NMR}$ measurements. X-Ray crystal data, powder X-ray diffraction data and Raman data before and after irradiation/heating cycles are now reported and provide consistent, unambiguous proof of the structural transformation.

3. In Figure S11, there are some flaws in the PXRD of coordination polymers. Thus the authors should re-measure their PXRD or perform the $^1\text{H-NMR}$ spectra to confirm the complete retro-conversion of these complexes.

Response: We thank the reviewer for their constructive criticism and have re-collected the PXRD patterns during the structural transformation. This data is provided in Figure S11 in the revised supplementary information. To expand on the previous PXRD data, we have also measured a second cycle of cyclisation/retroconversion and a third cyclisation to demonstrate the repeatability of the reversible structural transformation. With regards to quantifying the reversibility of our framework from **3** to **1** using heat, we have undertaken a further analysis of the *in situ* Raman spectra. Given the complete remission of the 1527 cm^{-1} peak in the forward photocyclisation reaction, we used this peak to quantify the percentage conversion over time. By deconvoluting the peak at 1527 cm^{-1} for the series of spectra collected in both the photo-irradiated and isothermal Raman measurements, we used the integrated area to define the percentage conversion from the fraction of the first (for the photo-irradiated Raman experiment) or last (for the isothermal Raman experiment) spectrum. The resulting percentage conversion for the retro-conversion reaction was 100%. Thus, we have now provided unambiguous evidence for the retro-conversion, and recyclability of the structural transformation.

4. Some interesting works related to the coordination polymers for single crystal-to-single crystal [2+2] photocyclisation should be cited. For examples: *Angew. Chem. Int. Ed.*, 2018, 57, 12696-12701; *Angew. Chem. Int. Ed.*, 2010, 49, 4767; *Chem. Commun.*, 2018, 54, 5831; *Chem. Commun.*, 2016, 52, 7990; *Inorg. Chem.*, 2018, 57, 849.

Response: We thank the reviewer for citing these interesting papers. The works have now been referenced and are cited throughout the revised introduction of the manuscript.

5. There are still some typo and grammatical mistakes in the manuscript which should be corrected and polished before its acceptance in this journal.

Response: We note the reviewer's concerns and have remedied any grammatical, linguistic and typographical errors throughout the manuscript.

Reviewer #2:

In this work, the authors report a reversible single crystal-to-single crystal double [2+2] cycloaddition in a metal-organic framework (MOF) which consists of electroactive tetrathiafulvalene (TTF) monomers. The SC-SC photocyclization has been reported in the past for several examples of MOFs and 2D polymer frameworks. But the examples of realizing the reversible photocyclization remain limited. Thus, the major novelty of this work as claimed by the authors lies in the first example of a MOF exhibiting a reversible SC-SC double [2+2] photocyclization based on the TTF core, which also leads to varying the electrochemical and optical properties. Basically, the novelty of the work can be fine if there are interesting properties demonstrated for the SC-SC photocyclization process in the newly synthesized MOFs. Nevertheless, there are substantial issues required to be addressed before the further consideration of this work for a possible publication in *Nature Communications*:

-There are too many claims such as, "the first example of..." and "a novel example of..." in the context. The

authors shall stay more neutral when describing major claims. I believe that every piece of work should be published as the first example of a specific research field.

Response: Agreed. We thank the reviewer for their constructive criticism and have reduced the number of claims accordingly. The introduction has now been revised to more clearly reflect precedents within the field.

-It is good that the authors use PXRD and in-situ Raman spectroscopy to demonstrate the reversibility of transition between **1** and **3**. Nevertheless, the missing key information of this work is about the quantification of the reversibility. From the latter results of square wave voltammetry and spectroelectrochemistry, it is clear that not all the monomers undergo the [2+2] photocyclization. Since the complete conversion of monomers can influence the intrinsic properties of the MOFs, the quantification of conversion and reversed ring opening reaction shall be demonstrated. Moreover, how many times have the authors repeated the reaction cycles between **1** and **3**?

Response: With regards to quantifying the reversibility of the framework retro-conversion from **3** to **1** using heat, we have undertaken a further analysis of the *in situ* Raman spectra. Given the complete remission of the 1527 cm⁻¹ peak in the forward photo-cyclisation reaction, we used this peak to quantify the percentage conversion over time. By deconvoluting the peak at 1527 cm⁻¹ for the series of spectra collected in both the photo-irradiated and isothermal Raman experiments, we used the integrated area to define the percentage conversion from the fraction of the first (for the photo-irradiated Raman experiment) or last (for the isothermal Raman experiment) spectrum. The resulting percentage conversion for the forward reaction was *ca.* 98.6% whilst the backward reaction achieved 100% conversion. The protocol that we have employed and the result of this analysis is presented in the ESI (Tables S4 and S5). The main text has also been edited to describe the quantification from the *in situ* Raman data.

With regards to the question of cyclability, we have updated Figure S11 to include PXRD data demonstrating two complete cycles and a third cyclisation (i.e., **1-3-1-3-1-3**). The data shows retention of crystallinity and further research will investigate the long-term stability of the material. In addition to the comment on monomer conversion influencing intrinsic properties, we have expanded our electrochemical analysis to include an intermediate electrochemical profile, and an adjusted profile for **3** that explains the effect that the degree of cyclisation has on the redox-active properties of the material (see Figure 3 and S15-16). The retro-converted sample was also measured to prove that the material's electrochemical profile returns to that observed for the as-synthesised sample of **1**.

-The authors claimed the modulation of electronic properties between **1** and **3**, but to be frank, I could not really find the electronic property studies in the context. The authors mainly describe the optical or photophysical studies of bulks samples of **1** and **3**. For the electronic property, I would expect some investigations on the electronic conductivity or so.

Response: We thank the reviewer for their comment. We suggest that the electronic properties do encompass electrochemistry and optical properties which are contactless techniques. The material would at best be semiconducting, and contact resistance issues that are common, within our experience, would mean that the results were tenuous. However, to avoid ambiguity or unsupported claims, we have removed references to electronic properties and rather, have specified electrochemical properties.

-The AFM investigation on MOF crystal phase change is interesting, but I am afraid that the connection to mechanical behavior is not really strong. It is more relevant for the volume change or height change of the crystals. For the mechanical behavior or study, I would expect something else, like the pressure, tensile strength, etc.

Response: We first would like to thank the reviewer for the positive comment. Strain and stress are the basic features of mechanical properties and the mechanical strain can be induced by many external stimuli such as electric/magnetic fields and temperature. The multi-functional materials stress the applications of mechanical strain generated by other factors. For example, due to the outstanding electromechanical coupling effects, piezoelectric materials are widely manufactured into the sensors and actuators. Similarly, the main focus of our research is the photoinduced strain rather than the traditional mechanical properties such as elastic stiffness and compliance.

Previously the photo-induced strain has been investigated on the inorganic ferroelectric materials such as LiNbO_3 and BiFeO_3 while their amplitudes, which are 0.04 % and 0.001 %, respectively, are quite small.^{1, 2} However, in our work, the photo-induced strain is originated from the phase transition, and the strain along [100] direction can attain as large as -11 %, which endows the materials with potential applications for photo-valve. Such a large deformation is also experimentally confirmed by the AFM techniques. The main idea of this work highlights the functionalities arise from the reversible single crystal-to-single crystal [2+2] photocyclization and reflects the energy transfer among different forms. Therefore, we thought the mechanical behaviours such as elastic stiffness, modulus the reviewer mentioned are beyond the scope of this work. Nevertheless, as this MOF sample contains several polymorphs, the elastic stiffness is expected to be different among different phases. We believe more following theoretical and experimental works will be conducted with respect to this phenomenon.

References:

1. Dingquan, X.; Jianguo, Z.; Shipin, Z.; Xiu, W.; Wen, Z.; Guoqin, L.; Guanfeng, X., Photostriction effect of doped LiNbO_3 single crystals. *Solid State Communications* **1991**, 79 (11), 1005-1007.
2. Kundys, B.; Viret, M.; Meny, C.; Da Costa, V.; Colson, D.; Doudin, B., Wavelength dependence of photoinduced deformation in BiFeO_3 . *Phys. Rev. B* **2012**, 85 (9), 092301.

-Some speculative sentences like in page 5 “We postulate that the double [2+2] photocycloaddition is possible....” shall be removed out.

Response: Agreed. Speculative comments have been removed from the manuscript.

-In the supplementary information, the reaction scheme for the synthesis of Py2TTF and the chemical structures of all intermediate compounds should be shown.

Response: Agreed. A synthesis scheme has been added as Scheme 1 in the supplementary information, along with chemical structures of all intermediate compounds.

Reviewer #3:

In their manuscript, D’Alessandro and coworker present a tetrathiafulvalene-containing metal-organic framework containing which undergoes [2+2] photocyclization.

The MOF and its phase changes were carefully analyzed by x-ray diffraction and Raman and optical spectroscopy. Changes of the redox potential were shown by cyclic voltammetry. Atomic force microscopy showed a shrinking of the crystal as result of the phase change due to light irradiation.

The data is well presented and comprehensible. The manuscript is well written.

Unfortunately, the field of light responsive MOFs is only partially cited and a classification of the present work is difficult. For instance, MOF single crystal-to-single crystal switching has been presented several times.

While the first MOF driven actuators have been published, here the potential application as actuator is given only as outlook. Likewise the electronic properties where MOFs with switchable conduction have

been already presented. Both features go beyond the here presented work. So, it remains vague how the presented MOF performs in comparison to the published MOF materials.

Although the presented MOF is certainly interesting from a scientific point of view, I cannot spot the major breakthrough expected for a publication in a high impact journal.

Due to the lack of novelty, I cannot recommend the publication in *Nat. Comm.* and suggest a more specialized journal.

Response: We thank the reviewer for their thoughtful comments and appreciate their perspective. This study was not intended to demonstrate a single and specific device or realised application. Rather, it aims to demonstrate a fundamental discovery, which we believe in this context is more significant when considering major breakthroughs as it provides an archetype to engender a new paradigm for photo- and thermally sensitive switchable redox-active MOFs. That discovery can be summarised as using double [2+2] photocyclization for the first time in a MOF to realise a range of new properties that can be controlled. Here, specifically, we explore a redox-active ligand and therefore there is potential for this easily photo-induced structural transition to control the electrochemical profile of the material. Importantly, we also demonstrate repeatable reversibility of this transition to enable a switching capability. The effects of this transition, however, extend beyond electrochemical activity and also affect a variety of other properties in the material, including the spectral features.

We recognise that there are a number of demonstrations of MOF applications including mention of their potential as actuators.^{1,2} The present study, however, envisions a multifunctional paradigm by designing a material that could be used simultaneously for many of these applications in the future. In this regard, we strongly believe that publication in *Nature Communications* is warranted, providing a platform for other researchers expert in applications to build upon our important fundamental scientific advance.

References:

1. Shi, Y.-X.; Chen, H.-H.; Zhang, W.-H.; Day, G.S.; Lang, J.-P., Photoinduced Nonlinear Contraction Behavior in Metal–Organic Frameworks, *Chem. Eur. J.* **2019**, *25*, 8543-8549.
2. Shi, Y.-X.; Zhang, W.-H.; Abrahamx, B.F.; Braunstein, P.; Lang, J., Fabrication of Photoactuators: Macroscopic Photomechanical Responses of Metal–Organic Frameworks to Irradiation by UV Light, *Angew. Chem. Int. Ed.* **2019**, *58*, 9453-9458.

REVIEWERS' COMMENTS:

Reviewer #1 (Remarks to the Author):

The authors have properly responded to the concerns raised by the three reviewers. I satisfied with all the changes they made and would like to recommend its publication in Nature Communications without further revisions.

Reviewer #2 (Remarks to the Author):

In the revision, the authors have mainly addressed my concerns. The quality of the work is mostly improved to my view.

One final remark: since Raman spectroscopy is not a qualitative technique for the evaluation of the conversion percentage, the authors shall put the accuracy in the discussion.

Response to referees comments for NCOMMS-19-11694A

We thank the two referees for their time in reconsidering our initial responses and for their additional comments and questions. Referee 2 only had raised an additional question.

Reviewer #2:

In the revision, the authors have mainly addressed my concerns. The quality of the work is mostly improved to my view. One final remark: since Raman spectroscopy is not a qualitative technique for the evaluation of the conversion percentage, the authors shall put the accuracy in the discussion.

Response: We thank the reviewer for their comment and acknowledge that our use of Raman spectroscopy to assess the extent of conversion and retro-conversion is only semi-quantitative. We have therefore considered the uncertainty limits for the method and have included the following discussion in the manuscript:

“Note that this method is semi-quantitative with an estimated uncertainty limit based on the accuracy with which band areas could be calculated of $\pm 5\%$.”

To ensure that readers are well-informed, we have also included this comment in the Supplementary Information with Supplementary Figures 4 and 5.